# Lurasidone use in Cannabis-Induced Psychosis: A Novel Therapeutic Strategy and Clinical Considerations in Four Cases Report

**DOI:** 10.3390/ijerph192316057

**Published:** 2022-11-30

**Authors:** Valerio Ricci, Giovanni Martinotti, Domenico De Berardis, Giuseppe Maina

**Affiliations:** 1San Luigi Gonzaga Hospital, University of Turin, 10043 Orbassano, Italy; 2Department of Neurosciences, Imaging and Clinical Sciences, Università Degli Studi G. D’Annunzio Chieti-Pescara, 66100 Chieti, Italy; 3National Health Service, Department of Mental Health, Psychiatric Service for Diagnosis and Treatment, Hospital “G. Mazzini”, 64100 Teramo, Italy; 4Department of Neurosciences “Rita Levi Montalcini”, University of Turin, 10043 Orbassano, Italy

**Keywords:** cannabis, tetrahydrocannabinol, first episode psychosis, lurasidone

## Abstract

Background: Lurasidone is an atypical antipsychotic approved for the acute and maintenance treatment of schizophrenia. Recently, lurasidone was also extended FDA approval for adults with major depressive episodes associated with bipolar I disorder (bipolar depression), as either a monotherapy or as adjunctive therapy with lithium or valproate. The use of low doses of atypical antipsychotics is an essential component of early intervention in psychosis, but little has yet been studied on first episode cannabis-induced psychosis. For its particular performance and tolerability, lurasidone is becoming an important option for the treatment of first-episode psychosis in youth. Case presentation four patients experiencing first cannabis-induced psychotic episode were treated with lurasidone. In all patients, there was an improvement in the clinical picture of psychosis. The recovery was positive, not only with the remission of positive and negative symptoms, but also regarding disruptive behaviour, with the return of functioning. All the patients were treated with lurasidone, with a target dose of 74–128 mg/day. No significant side effects were reported. Conclusion: There are non-controlled studies for the use of lurasidone in first episode psychosis cannabis induced. These findings suggest that lurasidone is an atypical antipsychotic beneficial in this clinical picture. Treatment with medium-high doses of lurasidone could be effective and tolerable in this phase of the disorder. Randomized control trials with longer follow-up are recommended to confirm these positive results.

## 1. Introduction

Cannabis is the most widely used illicit psychoactive substance; its widespread consumption is second only to alcohol and tobacco. Approximately 8–12% of regular cannabis users develop Cannabis Use Disorder (CUD) [1,2], defined by the DSM-5 as “a problematic pattern of cannabis use leading to clinically significant impairment or distress” (American Psychiatric Association, 2013) [3]. Among the 80 cannabinoids contained in cannabis, delta-9-tetrahydrocannabinol (THC) and cannabidiol (CBD) are the ones present in greater quantities. THC, the main psychoactive component, is a partial agonist of the cannabinoid receptors CB1 and CB2 and is believed to be responsible for the risk of developing psychotic symptoms [4].

Several investigations have been conducted to define the link between cannabis and first episode psychosis, with relevant results suggesting that cannabis may be an independent risk factor for psychotic onset [5,6,7,8,9]. The nature of the link between cannabis and psychosis, however, is not yet well understood, and although it is not yet possible to identify a single causal link, epidemiological evidence suggests a strong correlation between cannabis use and the risk of developing psychotic disorders [10,11]. Furthermore, cannabis accelerates the age of onset and worsens the course of psychotic illness [12,13]. Recent evidence shows that cannabis users have a two [10,11] to four [14] times greater risk of developing a psychotic disorder than non-users. Approximately one in every four individuals with schizophrenia has a concurrent diagnosis of CUD [15].

From a psychopathological perspective, several studies have found that cannabis use is related to earlier onset of psychotic illness, and a psychotic breakdown may occur almost three years earlier in cannabis users compared to non-users [16]. Cannabis use may influence the expression of prodromal symptoms and progression to psychosis in individuals at high risk, given the presence of subthreshold psychotic symptoms, genetic predisposition, or recent deterioration in global functioning [17]. Some studies have found more clinical severity, both in positive and negative symptoms, in cannabis users [18,19,20,21,22,23]. Moreover, cannabis-associated psychosis may be characterized by a greater number of days of hospitalization [24], a poor adherence to medication [25], lower pharmacological compliance [26], a higher relapse rate [27], and lower levels of functioning [28,29].

In these years, early intervention can not only delay the onset of psychotic illnesses, moderate their severity, and counteract their biological, psychological, and social consequences, but also play a crucial role in preventing deterioration of social functioning, even at the stage when psychosis is not yet manifest. Unfortunately, there exists no clearly established treatment strategy for first episode psychosis (FEP) patients presenting a comorbidity of substance use disorders, neither with cannabis. Clinical guidelines suggest that atypical antipsychotics should be administrated in FEP at low dose, with a potential increase depending on clinical results and the patient’s tolerability.

A new oral atypical antipsychotic, lurasidone, is approved by the Food and Drug Administration for the acute and long-term treatment of schizophrenia and bipolar I depression [30,31]. The pharmacodynamic profile of lurasidone shows antagonism for D2 dopaminergic receptors and serotoninergic 5-HT2A and 5-HT7 receptors, moderate partial agonist activity on the 5-HT1A receptor, and antagonism on α2c receptor subtypes. It has no affinity for M1 muscarinic and H1 histaminergic receptors, theoretically reducing adverse events such as weight gain, excessive sedation, and cognitive impairment [32]. Lurasidone has demonstrated efficacy in the treatment of bipolar depression, both as a monotherapy and as an adjunctive therapy with lithium or valproate [33,34]; therefore, some studies showed efficacy of lurasidone in major depressive disorders with mixed features [35].

This case series reports the efficacy of lurasidone in four patients who experienced cannabis-induced psychosis.

## 2. Case Reports

### 2.1. Case One

Mr A. is a 20-year-old male university student. Psychotic onset occurred about 12 weeks ago with clear positive symptomatology characterised by severe disorganised thought polarised on persecutory themes, auditory hallucinations, and insomnia. Clinical symptoms progressively worsened: the patient reported bodily transformation delusions and frequent auditory hallucinations that suggested he was hurting himself. Cannabis use began at the age of 16, first once a week, then once a day.

Mr A came to our emergency department with his sister describing psychotic symptoms. He voluntarily accepted admission to our Psychiatric department with the diagnosis of non-affective FEP.

Routine blood tests were performed without any significant alterations. Urine drug screening showed positivity for cannabis. Neuroradiological and cardiac examinations were normal. No history of medical and psychiatric comorbidities was reported. He had a family history of substance use disorder from both parents. He did not take any herbal medication. He was treated with lurasidone 37 mg, titrated to 74 mg after one week. He took lorazepam 2.5 mg twice a day, which was then tapered off for two weeks. After two weeks, he was discharged while maintaining lurasidone 74 mg/day. He showed significant improvement in disorganized behaviour and delusions. The total score of the Positive and Negative Syndrome Scale (PANSS) dropped from 88 (admission) to 47 (hospital discharge); positive subscale dropped from 30 to 10; negative subscale from 8 to 6. He had no side effects. He had a good response from pharmacological therapy with follow-up at 5 months. Beside pharmacotherapy, the patient conducted psychotherapeutic treatment. He no longer took cannabinoids and returned to academic activity.

### 2.2. Case Two

Mr C. is an unemployed 24-year-old man with a history of first episode psychosis two years ago. Prior to FEP, the patient suffered from retired life and an unusual interest in magic and oriental philosophy for at least a year. Afterwards, full-blown psychotic symptomatology was characterised by first rank symptoms, more specifically, somatic hallucinations and thought withdrawal. He developed increasing paranoid themes that his girlfriend had turned against him and that an assassin was sent to kill him. He had been regularly using cannabis since the age of 18. He had no family history of psychiatric disorders. He also had no significant medical history. Given the severe clinical condition, the patient was admitted to our psychiatric ward. Blood tests and urine drug screening were in the normal range, except for the cannabis screening being positive. After initial clinical assessment, he was treated with paliperidone 6 mg/day, with general clinical improvement and reduction of positive symptomatology, but with worsening of negative symptoms and a strong decreased motivation and sociality. PANSS total score dropped from 80 to 35; PANSS total score from 25 to 12; the negative scale scores remained high. After discharge, the patient continued to use cannabis, leading a reiterated life. Antipsychotic therapy was switched to aripiprazole 15 mg/day in an out-patient clinic, in an attempt to reduce negative symptoms. This pharmacological change did not produce any improvement in negative symptoms. After 6 months of treatment with aripiprazole, the psychopharmacological treatment was switched to lurasidone.

Lurasidone was titrated to 74 mg/die (starting 37 mg for two weeks). Clinical improvement appeared after three weeks, with disappearance of the avolition and abulia that were prevalent few months ago. One side effect was somnolence, which gradually disappeared after one month. He took flurazepam 15 mg/day at bedtime and delorazepam 3 mg daily to prevent potential withdrawal symptoms. These drugs were discontinued after one month. The patient continued taking lurasidone 74 mg/day. The PANSS negative subscale score diminished from 35 before starting treatment with lurasidone to 10 at one month of treatment. The patient also started attending group therapy with the aim of maintaining his sobriety.

### 2.3. Case Three

Miss K. is a 20-year-old female university student. Since the age of 17, she has regularly used cannabis (two joints a day). General functional decline and reduced executive function emerged 16 weeks ago: she presented disorganized behaviour, auditory hallucinations (threatening voices), and somatic-cenesthetic (a feeling that someone was pulling her shadow and that a control device influenced her movements). The symptoms worsened and she dropped out of university, expressing feelings of strong persecution. Her brother, worried by his sister’s behaviour, convinced her to go to the hospital. Thus, she was hospitalized in the department of Psychiatry with a diagnosis of FEP. The patient had a history of hyperthyroidism, medicated with methimazole 15 mg/day. Other than her mother suffering from depression, there was no psychiatric history in the family. Routine blood exams showed no significant abnormalities other than THC. On the first day of admission, the patient started lurasidone 37 mg/day that titrated to 74 mg/day after two weeks. Given the incomplete remission of positive symptoms, lurasidone was increased to 111 mg after one week, with good response.

The patient experienced a progressive remission of the psychotic symptoms, and after four weeks, she was discharged. PANSS total score dropped from 82 to 45, subscale positive scale from 29 to 7, and negative subscale from 12 to 4. The patient had been followed-up at the Local Mental Health Centre. Nine months after the start of the lurasidone treatment, the psychotic component appeared to be completely resolved, with no significant side effects. She also returned to her academic activity. The patient continues to take lurasidone at the maintenance dose of 74 mg.

### 2.4. Case Four

Mr T. is a 25-year-old male factory worker with no history of psychiatric illness. He was conducted by the police to our emergency room after seeing him wandering in a severe psychotic state on the edge of river.

On admission, he verbally threatened resident physicians and left the room abruptly. He presented with agitation and disruptive behaviour. He developed increasing paranoid delusions that his family had turned against him and that medical doctors wanted to kill him. He remained uncooperative with extreme agitation, confusion, and disorganization. Subsequently, intramuscular haloperidol 10 mg, and intravenous lorazepam 8 mg were administered, with partial response. The next day after admission, violent behaviour and paranoid delusions persisted. The patient appeared suspicious, irritable, and reactive to his psychiatrist. Haloperidol was stopped and he has been treated with Lurasidone a 128 mg/day (74 mg/day for two day; 128 mg mg/day afterward). He had tested positive for THC; other laboratory studies completed during his hospitalization were unremarkable. The patient began using marijuana at the age of 13. He had since used marijuana on a daily basis (two to three joints per day); he had, over time, gradually shifted to more potent forms of cannabis. After a few days, the patient confessed to having used “Spyce” for the past month, a highly concentrated form of THC, more potent than most recreationally available products.

One week after starting lurasidone, the patient was calmer and more compliant; psychotic symptoms gradually disappeared. After four weeks of treatment, he was discharged and referred to the nearby mental health centre. The patient only took delorazepam 2 mg/day for one week. The PANSS total score decreased from 100 to 40, PANSS positive scale decreased from 40 to 12, and the PANSS negative score was irrelevant. After 6 months, he showed improvement, with complete remission of psychotic symptoms and violent behaviour (lurasidone was reduced to 74 mg/daily as a maintenance dose). He returned to his job.

## 3. Discussion

The four case reports provide evidence of the effectiveness of lurasidone in FEP induced by cannabis. All patients showed improvement in both positive and negative symptoms and in global functioning. The clinical scenario differs significantly among patients: in patients one and three, positive symptoms decreased after two or four weeks of hospitalization. Patient two experienced improvement for negative symptoms incurred subsequently (after six months of previous antipsychotic therapy). Patient four presented with a disorganized, confused, and hostile behavior, and needed a longer hospitalization time. In all cases, lurasidone proved effective either as a first- or second-choice drug. All patients were not abusing other substances.

A critical diagnostic implication is clarified whether substance-induced psychosis represent an expression of intrinsic vulnerability or merely constitute a clinical expression of a more complex psychopathological dimension. The Diagnostic and Statistical Manual of Mental Disorders, fifth edition [3] defines substance-induced psychotic disorder as a “psychiatric illness characterized by delusions and/or hallucinations during or immediately following intoxication or withdrawal from substances”. The four factors leading to the development of psychotic symptoms in cannabis users are: THC/CBD ratio, frequency of use, quantity taken, age of intake (<25 years) [36]. For the first factor, more aggressive and positive symptoms in patients with FEP (patient n 4) are due to a high percentage of use of novel psychoactive substances, named Spice, a compound with one or more synthetic cannabinoids, which is much more harmful and unpredictable than cannabis. The development of induced psychosis may depend on frequency and quantity of cannabis use. It is clear that compounds with a higher THC concentration may be more likely to lead to the development of psychosis [10,12,37]. Starzer et al. showed that 32.2% of patients who suffered from substance-induced psychosis developed schizophrenia (26.0%) or bipolar disorder (8.4%) within the next five years [38]. Hence, psychotic symptoms may persist for a long time after the last intake of the substances. From a clinical point of view, substance-induced psychoses present more qualitative and quantitative alterations in consciousness (twilight state), more dissociative states, more frequent impairments of sensory perception (visual hallucination); Patients are often self-critical, with a higher degree of insight, and present violent and disruptive behavior [20,39].

Based on our observation, the clinical manifestation of FEP in cannabis users may be heterogeneous: psychotic symptoms can range from the classical psychotic signs to severe psychomotor agitation, disruptive, and dysphoric behavior [40].

To our knowledge, no clinical trials have been conducted in individuals with cannabis-induced psychosis, and literature is limited on medication treatment for individuals with schizophrenia and a co-occurring substance induced psychosis. After management of the acute agitation, when it is present, long-term treatment involves second generation antipsychotic (paliperidone, clozapine, risperidone, olanzapine, and quetiapine) [41,42] with regard to improvement, to distinct of psychopathological domains, reduce craving, and greater reduction of substance use compared with first generation antipsychotics. Clinical observations showed efficacy of aripiprazole in ultra high risk psychosis in individuals at the onset of psychosis [43], and cariprazine for treatment of methamphetamine use disorder [44,45].

Lurasidone is a second-generation antipsychotic that has long been authorized by the Food and Drug Administration (FDA) for the treatment of schizophrenia in adults, and for bipolar disorder, either as monotherapy or as combination therapy with lithium and/or valproic acid. The European Medicines Agency (EMA) gave a positive opinion for the marketing of the drug on January 2014, with the only indication of the treatment of schizophrenia [46]. The drug has been available in Italy since November 2017 with a dosage ranging from a minimum of 18.5 to maximum of 128 mg per day.

The efficacy of lurasidone in the acute treatment of schizophrenia was demonstrated in five designed multicenter studies (6-week duration, versus placebo or placebo and active treatment) conducted in subjects who met the criteria for schizophrenia (DSM-IV). The doses of lurasidone ranged from 37 to 148 mg (equivalent to 40 and 160 mg lurasidone hydrochloride), once daily [47,48,49,50,51]. Three trials have instead evaluated the long-term efficacy of lurasidone [52,53,54].

Based on the efficacy and safety observed from these studies, lurasidone becomes the first second-generation antipsychotic drug approved for the treatment of schizophrenia in youth 13–17 years old at doses of 40/80 mg/day.

To the best of our knowledge, no previous studies have been conducted regarding cannabis-induced psychosis treated with lurasidone. Together with the clear efficacy in treating positive symptoms, our clinical observations emphasize the efficacy on negative symptoms, as with patient 2, and on aggressive behavior, as with patient 4. Patients 1 and 3 received lurasidone as the first choice option, while for patients 2 and 4, lurasidone was prescribed after previous antipsychotic treatments. These results are in line with previous observations, in which lurasidone is an effective treatment option for both antipsychotic-naïve adolescents diagnosed with schizophrenia and adolescents previously treated with antipsychotic medication, on positive and negative symptoms [55,56,57], with greater efficacy in the drug-naive patient group than patients previously treated. Costamagna et al., noted an onset of improvement on lurasidone as early as week 1, and it was consistently observed from weeks 2 to 6 across the 40–160-mg/d dose range in 537 adolescents and young adults with schizophrenia in pooled post hoc analysis of double-blind, placebo-controlled 6-week studies [58]. Regarding aggressive behaviour, a previous study evaluated the efficacy of oral lurasidone for the treatment of mild to moderate levels of agitation in patients with schizophrenia. Treatment with lurasidone significantly reduced agitation as early as day 3 in the higher agitation subgroup, and by day 7 in patients in the lower agitation subgroup, with higher doses of lurasidone (120–160 mg/day) [59]. Our four patients showed that a moderate level of agitation, disruptive behaviour, and positive symptoms disappeared with higher doses of lurasidone.

This clinical experience confirms that lurasidone is a safe and well-tolerated drug, and no significant side effects were reported. Only one patient experienced sleepiness, which was easily resolved after one month. Lurasidone also appears to be effective in other symptom domains related to schizophrenia, such as depressive symptoms, which are often underestimated by clinicians [60].

There are some important limitations regarding these case reports; the first limitation concerns the time to intake: patient 2 initiated lurasidone after six months, while other patients initiated in the acute phase. The second limitation is that the follow-up was different between the four case reports.

## 4. Conclusions

The clinical case reports open the way for two reflections. The first reflection aimed to enrich the clinical heterogeneity of FEP in cannabis users (from sub-threshold, mild positive and negative symptoms). The second reflection pointed out the positive role of lurasidone.

To date, no clinical study has been conducted on the use of antipsychotics in cannabis-induced psychosis. However, the various above-mentioned clinical presentations enhance the insight and experience of the clinical use of lurasidone for varied expressions of cannabis-induced psychosis. Future studies are needed to further investigate the role of lurasidone for this category of patient.

## Data Availability

The data presented in this study are available on request from the corresponding author. The data are not publicly available due to privacy reason.

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
