# Peer review of "Lurasidone use in Cannabis-Induced Psychosis: A Novel Therapeutic Strategy and Clinical Considerations in Four Cases Report"

_ijerph, 2022, doi:10.3390/ijerph192316057_

Round 1

Reviewer 1 Report

Interesting and useful paper. I suggest a minor revision:

TITLE: I suggest that the authors modify their title to: “Lurasidone use in Cannabis-Induced Psychosis: .... "

ABSTRACT: Background: Lurasidone is not a very new antipsychotic. It has been available for many years (2010 in the U.S.). It is not approved for this indication in Europe, only U.S.

I suggest that the authors mention that RCTs on this topic are needed.

Keywords: Please remove the abbreviations from keywords.

INTRODUCTION

Please recheck the text because font sizes are different.

Please explain the abbreviation for FEP.

There are some critical reports on lurasidone use in bipolar disorders: “Comment on An Open Trial of Lurasidone as an Acute and Maintenance Adjunctive Treatment for Outpatients With Treatment-Resistant Bipolar Disorder. J Clin Psychopharmacol. 2016 Oct;36(5):520-1.”

Last sentence: efficacy or effectiveness?

CASE REPORTS

CASE 1

Did the authors check if this patient took any over-the-counter herbal medications?

Was this patient diagnosed with FEP?

CASE 2

Interesting that aripiprazole was used in this patient (because of its high potential for D2 receptor upregulation).

Why did the authors use two benzodiazepines together? They are not recommended for patients with an abuse/misuse history. Would they provide any treatment plan for the future?

CASE THREE (please use the numbers or words in all cases CASE 3)

Plan for the future (e.g., continuation/discontinuation)?

CASE 4

the police to = remove space

Aloperidole = HAloperidole?

Why did the authors use 128 mg/day of lurasidone? This dose was not proven to be better than 111 mg.

Did the patient receive only monotherapy?

DISCUSSION

I suggest that the authors mention the newest meta-analyses, including lurasidone.

Reviewer 2 Report

I read with great curiosity the article, which is a case report on the treatment of a cannabis-induced psychotic episode with the drug Lurasidone.
The article describes four patients experiencing their first cannabis-induced psychotic episode. The authors describe the dosage used and the level of symptoms as measured by the PANSS scale. The article contains a well-described introduction, clear case descriptions and a broad discussion. Merit-wise, the article is correct, requiring only a few editorial corrections:
-the titles of the described cases should be standardised. The authors use CASE 1 and then CASE FOUR
-the footnotes used are also incorrect - (18-19-20-21-22-23) is used instead of (18-23)

After correcting the editorial side of the manuscript, in my opinion it qualifies for publication 
